# *Toxocara* species environmental contamination of public spaces in New York City

**Donna L. Tyungu**[1,2,3,4]*, **David McCormick**[2], **Carla Lee Lau**[4], **Michael Chang**[3], **James R. Murphy**[3], **Peter J. Hotez**[2], **Rojelio Mejia**[2☯]*, **Henry Pollack**[4☯]

**1** Department of Pediatrics, Section of Infectious Diseases, The University of Oklahoma Health Sciences Center, Oklahoma City, Oklahoma, United States of America, **2** Department of Pediatrics, National School of Tropical Medicine, Baylor College of Medicine, Houston, Texas, United States of America, **3** Department of Pediatrics, Pediatric Infectious Diseases, McGovern Medical School, the University of Texas Health Science Center at Houston, Houston, Texas, United States of America, **4** New York University, NYU Langone, Department of Pediatrics, New York, New York, United States of America

☯ These authors contributed equally to this work.
* donna-tyungu@ouhsc.edu (DLT); rmejia@bcm.edu (RM)

## Abstract

Human toxocariasis has been identified as an under-diagnosed parasitic zoonosis and health disparity of significant public health importance in the United States due to its high seropositivity among socioeconomically disadvantaged groups, and possible links to cognitive and developmental delays. Through microscopy and quantitative PCR, we detected that *Toxocara* eggs are widespread in New York City public spaces, with evidence of significant levels of contamination in all five boroughs. The Bronx had the highest contamination rate (66.7%), while Manhattan had the lowest contamination rate (29.6%). Moreover, infective eggs were only found in the Bronx playgrounds, with over 70% of eggs recovered in embryonic form and the highest egg burden (p = 0.0365). All other boroughs had eggs in the pre-infectious, unembronyated form. *Toxocara cati*, the cat roundworm, was the predominant species. These results suggest that feral or untreated cats in New York City represent a significant source of environmental contamination. These findings indicate that human toxocariasis has emerged as an important health disparity in New York City, with ongoing risk of acquiring *Toxocara* infection in public spaces, especially in poorer neighborhoods. There is a need for reducing environmental *Toxocara* contamination. Additional rigorous public health interventions should explore further approaches to interrupt transmission to humans.

## Author summary

*Toxocara canis* and *Toxocara cati* are helminth worms that infect dogs and cats, respectively. Infected dogs and cats will defecate thousands of *Toxocara* eggs into the environment. Humans are incidental hosts and are exposed when consuming contaminated soils via the fecal-oral route. After leaving the gastrointestinal tract, the *Toxocara* larvae will enter the vasculature and can migrate to any major organ system, including lungs, ocular,

**Data Availability Statement:** All relevant data are within the manuscript and its Supporting Information files.

**Funding:** Acknowledgemets: This research was supported in part by the Committee of Interns and Residents Patient Care Trust Fund Research Project Grant funded from 2015-2017. Research funding support for RM was provided by the U.S. Department of Health and Human Services, Health Resources and Services Administration for Baylor College of Medicine Center of Excellence in Health Equity, Training and Research (Grant No: D34HP31024). Funding also provided by the Texas Children's Hospital Center for Vaccine Development, and the National School of Tropical Medicine, Baylor College of Medicine. The funders had no role in study design, data collection and analysis, decision to publish, or preparation of the manuscript.

**Competing interests:** The authors have declared that no competing interests exist.

and central nervous system. Symptoms can range from mild muscle aches to severe asthma, blindness, and encephalitis. Humans are not definitive hosts of the parasite and cannot transmit *Toxocara* eggs to the environment or other humans. There is a need for research on the sanitary impact of *Toxocara* for both humans and animals, especially in large urban cities such as New York City. Poverty is also associated with higher rates of toxocariasis, with more contamination in poorer neighborhoods where animal control, deworming of pets, and less sanitary conditions exist. This study aims to understand further the disparity of lower socioeconomic areas having higher rates of contaminated parks and playgrounds, comparing the five boroughs of New York City.

## Introduction

*Toxocara canis* and *Toxocara cati* are ascarid nematodes that ubiquitously infect dogs and cats and can result in environmental contamination if the feces of infected animals contaminate community spaces. Eggs deposited in the soil can exhibit cryptobiosis when environmental conditions are not ideal and may survive for many years. [1] Human ingestion of embryonated eggs through contaminated soil, poor hygiene practices, or uncleaned vegetables, can result in paratenic zoonotic toxocariasis. [2–4] As *Toxocara* eggs develop in the soil, it is possible to detect the developmental progression of the helminth from germinal cells to the presence of viable infective larvae via microscopy. Eggs containing fully developed larvae are infectious to humans, whereas *Toxocara* at earlier stages of development are pre-infectious and cannot lead to toxocariasis. Following the ingestion of an embryonated egg, the third stage larva enters the bloodstream. It burrows through body tissues, where the worms can accumulate in the eye, brain, liver, or skin, leading to visceral or ocular larva migrans, blindness, subclinical cerebral infection, or covert infection which can diminish neurological cognition or result in developmental delays. [5–9]

Toxocariasis is an underreported and understudied disease in the United States. Both the parasite and human toxocariasis have been described as 'enigmatic' due to numerous deficiencies in the understanding of the organism, including the role of cerebral toxocariasis and the possible link to neurocognitive deficits and blindness in children. [2, 5] The US Centers for Disease Control and Prevention (CDC) identifies human toxocariasis as one of five neglected parasitic infections in the U.S., and it is possibly the most common helminthiasis in the United States after pinworm (*Enterobius vermicularis*). [8, 10] Toxocariasis is a neglected disease of poverty due to its disproportionately high seroprevalence in areas of community poverty, especially among underrepresented minority populations living in poor areas. [10–12]

Among the most enigmatic features of toxocariasis is its covert form, which generally is not associated with visceral larva migrans. Covert toxocariasis is subclinical, with only eosinophilia as a biomarker for suspicion of infection. [4] Covert toxocariasis has been linked to cognitive and developmental delays, lung dysfunction, and asthma. However, research is still in the nascent stages of understanding the full clinical spectrum of illness caused by *T. canis* and *T. cati*. [5] Overall, toxocariasis may be severely underdiagnosed due to the covert nature of the illness and gaps in medical knowledge; it is a disease that should not be overlooked as a cause of neurocognitive delay in children. [2, 4, 5, 7, 13, 14] A 1987 study in New York City (NYC) correlated *T. canis* seroprevalence to neurocognitive deficits in children. [5, 15, 16] Prior research has also shown that simple incubation of a *Toxocara* embryo allows it to become infective, and after infection, living larvae have been found in the brains of mice. [1] Mice with infected brains have significant impairment, including cellular damage, and may have different

areas of brain involvement depending on the infecting *Toxocara* species. [17, 18] Covert toxo-cariasis may also represent an important environmental cause of asthma among disadvantaged American children. [19, 20]

Throughout the history of the National Health and Nutrition Examination Survey (NHANES), the prevalence of *Toxocara* within the United States population has varied, rang-ing from 5%-14% during analyses from 1988–1994 and 2011–2014. [11, 21] In both studies, *Toxocara* seropositivity was disproportionately higher in persons from lower socioeconomic status communities, particularly in Hispanic, non-Hispanic black communities, and those with less than a high-school education. [11, 21]

Few studies have been conducted in the United States, and most were published over two decades ago. [22–28] In the few US studies, contamination rates ranged from 0.3%-27.5%, with the contamination rate defined as the number of eggs found per park or space evaluated. [22–28]

Walsh et al. created a predicted probability model to estimate the seroprevalence of toxocar-iasis in NYC using estimates from the NHANES III and correlated findings with available sociodemographic information, which suggested higher levels of exposure to *Toxocara* species in low income, minority, and immigrant communities. [29] This study is the first geo-surveil-lance of Toxocara species in all five boroughs of NYC attempting to address the potential risk of infection within NYC communities by first examining soil from around the city.

## Methods

### Sample collection

Samples were collected from 3–5 cm below the surface after the top layer of soil was removed. Five samples were collected from each location on the same day, and each sample was from a different geographic area of the sample site, attempts were made to sample the entire site equally. These were combined before laboratory analysis into one larger sample weighing 100-150g. The samples were collected randomly during October 2015-July 2016. A sample size cal-culation was used to determine the level of precision using a metanalysis (Fakhri et al.) confi-dence interval of 16 to 27% soil contamination worldwide from 42,797 samples, 79 samples were needed for statistical significance, and a total of 91 samples were collected for this study. [30] A total of 91 sites were surveyed in New York City from October 2015-July 2016, and samples were collected from different sites at different times during this period. Of these, 27 (29.6%) were in Manhattan, 15 (16.5%) were in the Bronx, 13 (14.3) were in Brooklyn, 18 (19.8%) were in Queens, and 18 (19.8%) were in Staten Island. Sample sites were selected at random to maximize coverage of each borough, due to limited access, parts of Brooklyn and Staten Island were not tested.

**Microscopic analysis.**   Samples were analyzed using a modified soil flotation technique derived from previously published floatation methods of helminth egg recovery from the soil and human stool. [31, 32] Sodium nitrate was chosen as the optimal flotation solution due to elevated specific gravity (1.25–1.35), best-published egg recovery results, and low laboratory cost. All specimens were sifted to remove large debris before being washed in 0.1% Tween 20 three times. Samples were subsequently mixed with approximately 10 to 14 ml sodium nitrate, specific gravity 1.30 (Vedco Feca Med, St Joseph, MO) enough to form a convex meniscus at the opening of the tube. With a coverslip in place, samples were centrifuged at 500 g for 5 min-utes and allowed to stand for 5 minutes before viewing using a standard light microscope. Eggs, if present, were easily visualized and counted using this technique. Parasites were identi-fied by structural characteristics, and embryonated forms were described if larvae were noted within the egg. Other incidentally discovered parasites were recorded and categorized using a

**Table 1. Prevalence of *Toxocara* sp. including the presence of larval forms in specimens collected from sites in each NYC borough.** Larvated, infectious *Toxocara* sp. were only found in the Bronx, the borough that had the highest prevalence of Toxocara contamination.

| Borough | Positives/Total | % Toxocara positive (95% CI) | Eggs (mean and range) | Larvated forms |
|---|---|---|---|---|
| Bronx | 10/15 | 66.7 (41.7–84.8) | 15.7 (1–102) | +++ |
| Staten Island | 7/18 | 38.9 (20.3–61.4) | 3.4 (1–5) | - |
| Queens | 6/18 | 33.3 (16.3–56.3) | 1.8 (1–2) | - |
| Brooklyn | 4/13 | 30.8 (12.7–57.6) | 2.7 (1–7) | - |
| Manhattan | 8/27 | 29.6 (15.9–48.5) | 6.3 (4–12) | - |

subjective semiquantitative "+" system (Table 1). Coverslips were then washed into 2mL Eppendorf tubes with DNase-free water and saved at -20 degrees Celsius for later real-time PCR testing.

**Molecular methods.** Parasite DNA was extracted from soil samples with MP FastDNA spin kits for soil (MP Biomedical, Santa Ana, CA) using supernatant from the coverslips from the previous centrifugation step. We processed this supernatant using bead beating for five minutes at 3,000 RPM, followed by a ten-minute heating step at 90°C to promote egg disruption. This same technique was utilized to extract DNA from *T. canis* and *T. cati* eggs for the standard controls. [33] The *T. canis* and *T. cati* DNA from these samples were quantified using a multi-parallel real-time quantitative polymerase chain reaction (qPCR) protocol. [33] All samples underwent qPCR testing using previously published probes with the following modifications [*T. canis* (5'-FAM-CCATTACCACACCAGCATAGCTCACCGA-3'-NFQ-MGB) and *T. cati* (5-FAM-TCTTTCGCAACGTGCATTCGGTGA-3'-NFQ-MGB)] and forward primers [*T. canis* (5'-GCGCCAATTTATGGAATGTGAT-3') and *T. cati* (5'-ACGCGTACGTATGGA ATGTGCT-3')] and shared reverse primer 5'-GAGCAAACGACAGCSATTTCTT-3') to both *Toxocara* species. [34] PCR was performed using an ABI ViiA 7 (Applied Biosystems, Foster City, CA) as previously described [33]. A sample was considered positive if there was detectable DNA at or before a cycle threshold of 38. The threshold of positivity was determined by *T. canis/cati* genomic DNA dilutions for the dynamic range. *T. canis* and *T. cati* eggs were isolated and washed with distilled water three times. Eggs were counted using a McMasters microscope slide (Advanced Equine Products, Issaquah, WA), and serial dilutions were used as standards. All samples were performed in duplicate and were repeated for discordant results. A standard curve was constructed using serial dilutions of 10,000 *T. cani and T. cati* eggs. Standards were used as positive controls with no-template for negative controls. A known concentration of plasmid internal control was added to all samples prior to bead beating to validate successful DNA extraction [35].

## Demographic data

Socioeconomic data were obtained from the websites of the US Census and the New York State Department of health. [36, 37]. Zipcodes were recorded from each sampling area, and New York City income data per zip code was obtained from a website of current census bureau income statistics for United States zip codes. [38]

## Statistical methods

Statistical analysis was carried out using GraphPad Prism version 7.0d (GraphPad, La Jolla, CA). The number of *Toxocara* eggs was compared by zip code, and *Toxocara* eggs per sample site were compared to individual boroughs using the Kruskal-Wallis test. Comparing the prevalence of *Toxocara* species by zip code or borough, a spearman rank was conducted comparing

## Results

### Environmental survey results (Microscopy results)

Of the 27 Manhattan sample sites, *Toxocara* eggs were found in 29.6% of playgrounds with at least one egg being identified by microscopy. Of the Staten Island sites tested, 38.9% had *Toxocara* eggs found, of the 13 Brooklyn sites tested 30.8% had eggs found, Queens had 33.3% of sites contaminated, and the Bronx had 66.7% sites contaminated (Table 1). The overall prevalence of 38.5% (35/91) contaminated playgrounds in NYC.

Infective embryonated eggs (Fig 1A and 1B, Fig 2) were only found in the Bronx playgrounds, with over 70% of eggs recovered being in their larvated state. All other boroughs had eggs in the pre-infectious, unembryonated (unlarvated) form (Fig 1C, Fig 2).

### *Toxocara* species results (qPCR results)

Of all the positive samples, six were found to have *T. cati* DNA present (Fig 3, Table 1). None of the parks were found to have *T. canis* DNA present. More than 4 of 6 parks that tested positive were from samples with the highest number of larvated eggs, accounting for the higher copy numbers of genomic DNA present in many of the samples (mean 19.5, 0 to 102 eggs). (Fig 3) The six positive samples had calculated egg counts from the standard dilutions using linear regression that correlated to 40 –Cycle threshold (Ct) (Spearman r = 0.999, p < 0.0001). One sample (Bk8) was negative by microscopy but positive by qPCR (13.958, 40-Ct). One egg of either *T. canis* or *T. cati* in 500 μl of water was consistently detected by qPCR with Ct values of 37 to 38.

### Association of geo-contamination and socioeconomic data

Contamination was more prevalent in lower socioeconomic neighborhoods. Manhattan, the borough with the highest median income ($97,255) of zip codes where sampling was performed, had the lowest rate of contamination (29.6%) compared to a contamination rate of 66.7% in the Bronx, which had the lowest median income ($26,131) (Fig 4). Queens, Brooklyn, and Staten Island had contamination rates from 30.8–38.9% and median incomes between $57,452-$69,998, Spearman r = -0.7, p = 0.233 (Fig 4). The Bronx sample sites had the highest mean egg burden (Bronx: 16.4, Queens: 1.8, Brooklyn: 1.9, Staten Island: 3.0, Manhattan: 5.9, p = 0.0365) and prevalence. (Fig 5) Overall, there was an inverse relationship between median income and the likelihood of *Toxocara* species contamination (Fig 5).

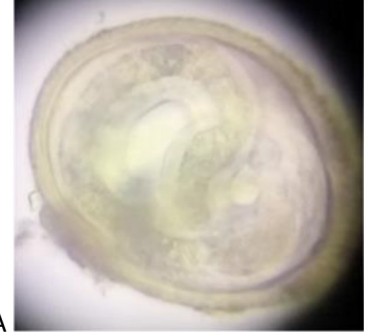 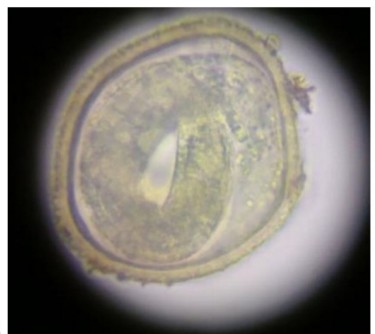 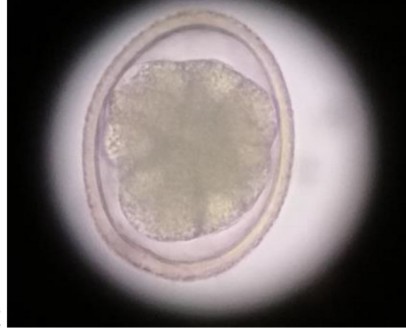

**Fig 1.** *Toxocara* forms isolated in NYC specimens: a and b: Larvated 'infective' *Toxocara* eggs at 40x magnification, c: Unlarvated *Toxocara* egg at 40 x magnification, obtained in NYC study.

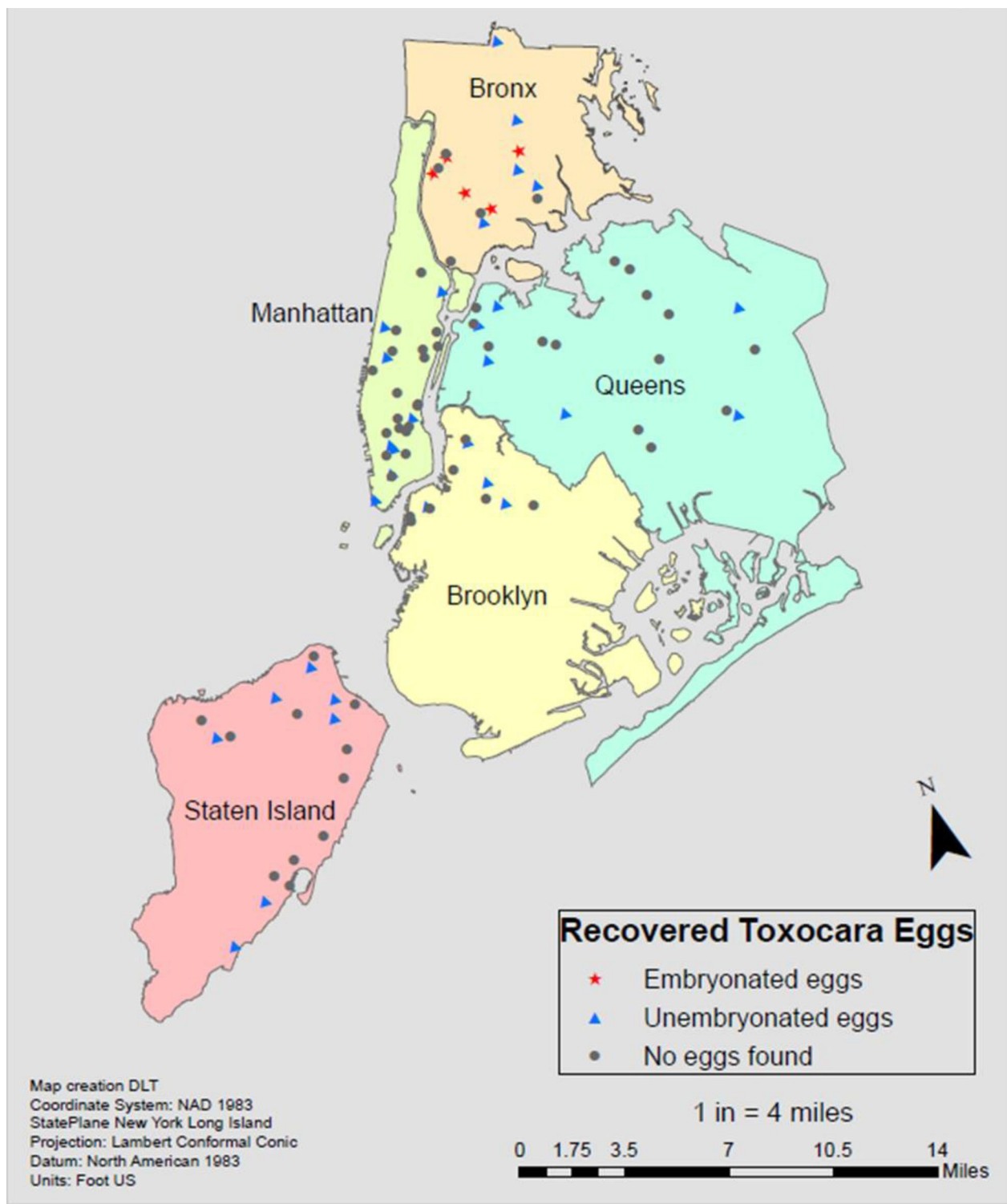

**Fig 2. Geospatial results of *Toxocara* sampling across NYC.** *Toxocara* was found in those sites represented by either blue triangles (unembryonated egg(s)) red stars (embryonated egg(s)). *Toxocara* was not found at sites marked by black circles. The highest density of *Toxocara* geo-contamination was found in the Bronx, which was the only borough where infected larvae were found. Sampling ArcMap of NYC, created using ArcGIS 10.6.1.

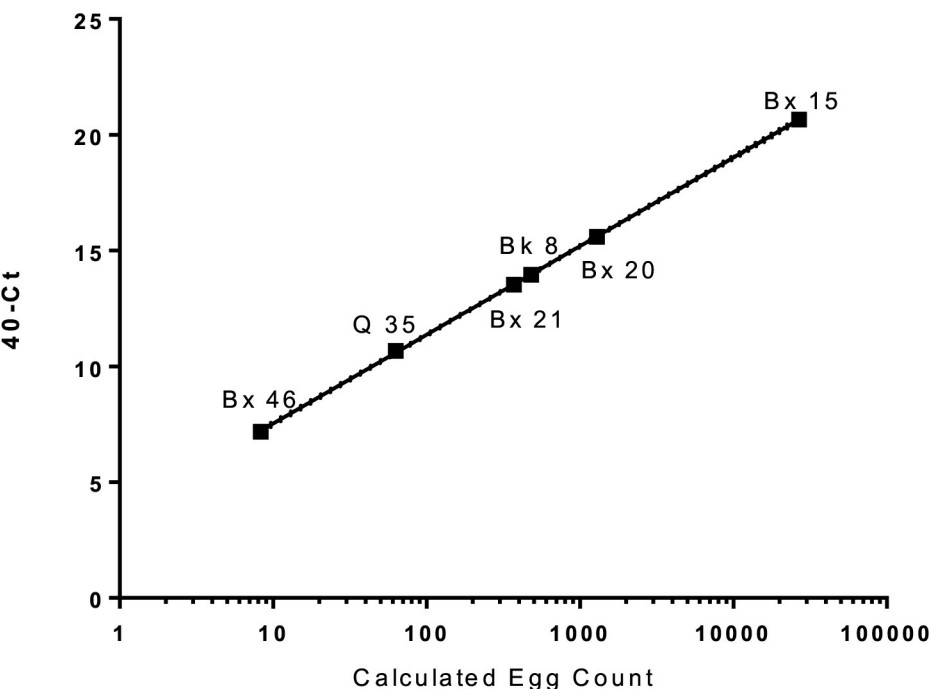

**Fig 3. qPCR results, parks that were qPCR-positive were contaminated with *T. cati*.** Calculated egg count was derived from dilutions of a known amount of *Toxocara cati* eggs and correlated to 40 –Ct (Spearman *r* = 0.999, p < 0.0001).

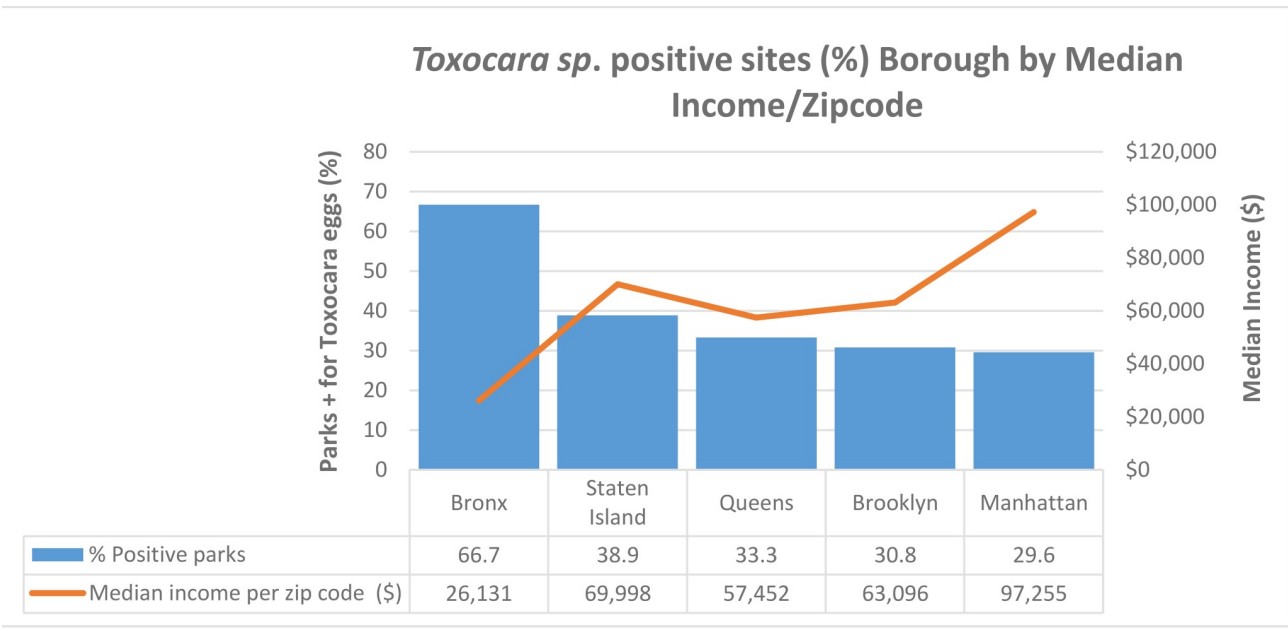

**Fig 4. Association of positive *Toxocara* results with the median income of residents living in the ZIP codes of sampled sites.** The line represents the median income of families living in the zip code where *Toxocara* testing was performed. Results aggregated by borough. (Spearman r = -0.7, p = 0.233).

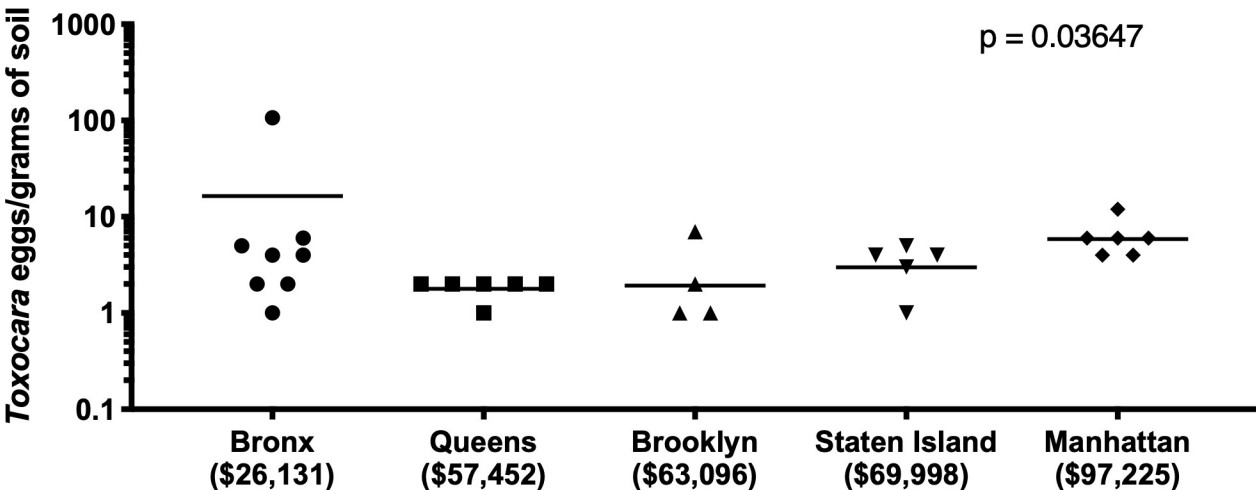

**Fig 5. *Toxocara* eggs/park by borough.** Kruskal-Wallis P = 0.0365. Eggs were not equally distributed in the parks by borough. The Bronx* has the highest egg burden compared to the other boroughs.

## Discussion

In this first environmental study of *Toxocara* prevalence in NYC, approximately one-third of sites sampled throughout the five boroughs contained *Toxocara* eggs (Table 1). The distribution of contamination was not equal across the city. The Bronx had the highest *Toxocara* burden and prevalence, compared to the other boroughs (Figs 4 and 5). All boroughs, except the Bronx, had eggs in their unlarvated, pre-infectious form (Fig 1).

From the qPCR data, *T. cati* was the only species identified. This suggests that cats are the predominant reservoir causing contamination in New York City play areas, although dogs can transiently carry *T. cati* from eating cat feces but are not part of the parasites' life cycle. [39] In fact, the finding of cat contamination is not entirely unprecedented, as NHANES data have shown high co-infection rates with toxocariasis and toxoplasmosis nationally, suggesting that cats may represent an important but often underappreciated source of *Toxocara* eggs. [40] There are several reasons for predominant feline contamination. Park fencing consists of vertical metal bars, which will effectively prevent dog entry into parks (unless pet owners allow access) but are unlikely to restrict stray cat access. Furthermore, people are more likely to remove feces from their dogs than cats. Cats can be infected with *T. cati* through all stages of life, whereas dogs are most severely infected as puppies. [41] Veterinary data in the New York City area shows that cats test positive for roundworm infections at least four times more frequently than dogs, regardless of the borough. [42] Furthermore, pet cats shed *Toxocara* more commonly than pet dogs. [43] Although the samples were not optimally processed for qPCR and the lack of *T. cani* DNA may be attributed to the small sample size.

Similar results have been recently reported in several cities in Europe [44, 45] and have important implications in determining strategies to reduce contamination and risk for human infection. [39] In a recent metanalysis by Fakhri et al., the average soil contamination with *Toxocara* in the USA was 4 to 23% playgrounds in NYC likely are have concentrations of animal exposure and results in higher egg counts.[30] Several considerations exist regarding why *Toxocara* qPCR was discordant for samples that were positive or negative by microscopy. One may be due to the way the specimens were saved and prepared for qPCR. The qPCR is not a direct comparison to microscopy as samples were first processed for microscopic analysis, and qPCR was performed on the supernatant obtained from the coverslips of the slides. The

specific transfer method after microscopy could account for a loss of eggs. Quantitative PCR was performed on the supernatants removed from the slide subsequent to the microscopy examination and not directly from the soil specimen. Given the low number of eggs (1–2) in many specimens, any loss could result in a negative qPCR. Specimens with embryonated eggs have an abundance of *Toxocara* DNA, often with very high calculated genomic equivalents, while non-larvated cells have only a single copy of DNA. Thus, PCR may better be able to identify samples with larvated cells present. Sample Bx15 was outside the dynamic range of the standards, and this is likely because several eggs were in the larvated stage and would have many copies of the target sequence, giving an exaggerated calculated egg count (Fig 3). Sample Bk8 was negative by microscopy, but positive by qPCR with significant 40-Ct value (13.958), showing that microscopy is subjective and can miss visualizing eggs. Prior literature studies [46, 47] suggest that these eggs can adhere to plastic and may have been lost during the rinsing of the coverslip and slide when the eggs were transferred into the Eppendorf tube in preparation for freezing. Other limitations are samples were collected in different seasons, with changes in environmental temperatures influencing embryonated eggs. These temperature changes could have influenced the larvated egg results found throughout the boroughs. Geography can also influence the prevalence, burden, and stages of embryonated eggs in the environment. Staten Island and Manhattan are not contiguous with the mainland and can have decreased migrating dogs or cats, therefore impacting the results.

The distribution of *Toxocara* contaminated parks was not homogenous across the city but was more prevalent in areas of lower socioeconomic status. Of the five boroughs, the Bronx has the lowest median income but the highest level of soil contamination with the highest number of infectious, larvated parasites. Based on embryonated/larvated eggs, the risk of Toxocariasis from soil ingestion was highest in the Bronx. It is unknown whether these areas have more pets or stray cats, but given the association with the lower-income, it is likely that the higher contamination rate and the higher infectious potential may be related to the ability to pay for regular veterinary check-ups and deworming of pets. Animals that frequent contaminated areas are likely to become re-infected even after deworming if neighboring animals have not been dewormed. Brooklyn, Queens, and Staten Island have higher median incomes with 30–38% geo-contamination. Manhattan has the highest median income but the lower percentage of contamination at 29.6%, but still indicating that over one-quarter of Manhattan sites tested have *Toxocara* present. Because there was variability in the soil weights between playgrounds (100 gm to 150 gm), there is a potential for sampling bias, since increased grams of soil can mean more eggs available to be seen by microscopy or detected by qPCR. This discrepancy may have increased the number of eggs in the Bronx (Fig 5). The overall high level of parasites throughout all boroughs may be a result of the 'pet boom'; the number of household U.S. cats and dogs has more than doubled in the past four decades, contributing to an increasing problem of stray cats in poorer neighborhoods. [48]

A disparity of higher *Toxocara* environmental contamination distribution in poorer neighborhoods is present and likely translates into an actual health disparity, but this remains to be evaluated as there are other ways that *Toxocara* can be transmitted to humans. Recent definitive data on the seroprevalence of *Toxocara* infection in children and adults living in those areas are lacking. However, results from a 1987 study did find higher seroprevalence in children and adolescents from poorer neighborhoods in NYC. [15] Furthermore, both *Toxocara* NHANES studies suggest minority individuals and those with lower socioeconomic status are at greater risk of Toxocariasis. [11, 21] Therefore, the prudent approach, given the potentially serious health consequences of *Toxocara* infection, especially for young children and adolescents, is to assume that transmission to humans will occur and to make every effort to reduce environmental contamination by *Toxocara* as much as possible and as soon as possible.

Beyond its importance as a human health disparity, *Toxocara* soil contamination is a One Health issue; it impacts the cleanliness of the environment and may impact the health of domestic pets and wildlife. Infective eggs represent an environmental risk and potential health hazard to children with pica. Other paratenic hosts of this parasite include rodents, birds, or rabbits, which, when infected, continue to contribute to the lifecycle of the parasite. Finally, other larvated parasites were seen by microscopy but unable to be definitively identified in these samples, posing a potential infectious risk to visiting animals or children.

## Conclusions

*Toxocara* species are common pet parasites that are found in the sand and soil of almost one-third of sample sites in NYC, especially in poorer neighborhoods. The predominant species in NYC appears to be *T. cati*. Preventive measures should be taken. These include improved fencing of play areas to prevent feral cat entry, deworming of domestic pets according to national veterinarian guidelines, better control of stray cats and dogs, picking up feces of pets, avoiding consumption of food that may have become contaminated, restriction of children with pica from play areas, and frequent handwashing after visiting play areas before ingestion of food or snacks.

## Acknowledgments

The authors thank all current and past collaborators for their contributions to *Toxocara* research and acknowledge the inability to site all peer-reviewed publications. DLT is grateful to those who have helped with advice and education in regard to this research, especially Susan Little DVM Ph.D. DACVM, Yonghua Li MD, William Borkowski MD, Mona Rigaud MD, Celia Holland BSc Ph.D., and Gloria Heresi MD. Special thanks to Dwight Bowman Ph. D. for providing *Toxocara cani* eggs for qPCR validation, and Anne Zajac DVM MS Ph.D. for providing feline samples infected with *Toxocara cati* eggs for qPCR validation.

## Author Contributions

**Conceptualization:** Donna L. Tyungu, Rojelio Mejia, Henry Pollack.

**Data curation:** Donna L. Tyungu, David McCormick, Rojelio Mejia, Henry Pollack.

**Formal analysis:** Donna L. Tyungu, Rojelio Mejia, Henry Pollack.

**Investigation:** Donna L. Tyungu, David McCormick, Rojelio Mejia, Henry Pollack.

**Methodology:** Donna L. Tyungu, David McCormick, Rojelio Mejia.

**Supervision:** Rojelio Mejia, Henry Pollack.

**Validation:** Rojelio Mejia, Henry Pollack.

**Writing – original draft:** Donna L. Tyungu, Rojelio Mejia, Henry Pollack.

**Writing – review & editing:** Donna L. Tyungu, David McCormick, Carla Lee Lau, Michael Chang, James R. Murphy, Peter J. Hotez, Rojelio Mejia, Henry Pollack.

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
