## [Decision Letter · Decision Letter 0]

12 Nov 2019

Dear Dr Mejia,

Thank you very much for submitting your manuscript "Toxocara species environmental contamination in public spaces in New York City" (#PNTD-D-19-01601) for review by PLOS Neglected Tropical Diseases. Your manuscript was fully evaluated at the editorial level and by independent peer reviewers. The reviewers appreciated the attention to an important problem, but raised some substantial concerns about the manuscript as it currently stands. These issues must be addressed before we would be willing to consider a revised version of your study. We cannot, of course, promise publication at that time.

We therefore ask you to modify the manuscript according to the review recommendations before we can consider your manuscript for acceptance. Your revisions should address the specific points made by each reviewer. 

When you are ready to resubmit, please be prepared to upload the following:

(1) A letter containing a detailed list of your responses to the review comments and a description of the changes you have made in the manuscript.

(2) Two versions of the manuscript: one with either highlights or tracked changes denoting where the text has been changed (uploaded as a "Revised Article with Changes Highlighted" file); the other a clean version (uploaded as the article file).

(3) If available, a striking still image (a new image if one is available or an existing one from within your manuscript). If your manuscript is accepted for publication, this image may be featured on our website. Images should ideally be high resolution, eye-catching, single panel images; where one is available, please use 'add file' at the time of resubmission and select 'striking image' as the file type. 

Please provide a short caption, including credits, uploaded as a separate "Other" file. If your image is from someone other than yourself, please ensure that the artist has read and agreed to the terms and conditions of the Creative Commons Attribution License at http://journals.plos.org/plosntds/s/content-license (NOTE: we cannot publish copyrighted images). 

(4) If applicable, we encourage you to add a list of accession numbers/ID numbers for genes and proteins mentioned in the text (these should be listed as a paragraph at the end of the manuscript). You can supply accession numbers for any database, so long as the database is publicly accessible and stable. Examples include LocusLink and SwissProt.

(5) To enhance the reproducibility of your results, we recommend that you deposit your laboratory protocols in protocols.io, where a protocol can be assigned its own identifier (DOI) such that it can be cited independently in the future. For instructions see http://journals.plos.org/plosntds/s/submission-guidelines#loc-methods

While revising your submission, please upload your figure files to the Preflight Analysis and Conversion Engine (PACE) digital diagnostic tool, https://pacev2.apexcovantage.com/ PACE helps ensure that figures meet PLOS requirements. To use PACE, you must first register as a user. Then, login and navigate to the UPLOAD tab, where you will find detailed instructions on how to use the tool. If you encounter any issues or have any questions when using PACE, please email us at figures@plos.org.

We hope to receive your revised manuscript by Jan 11 2020 11:59PM. If you anticipate any delay in its return, we ask that you let us know the expected resubmission date by replying to this email.

To submit a revision, go to https://www.editorialmanager.com/pntd/ and log in as an Author. You will see a menu item call Submission Needing Revision. You will find your submission record there. 

Sincerely,

Celia Holland

Guest Editor

Christine Budke

Deputy Editor

Reviewer's Responses to Questions

**Key Review Criteria Required for Acceptance?**

**Methods**

-Are the objectives of the study clearly articulated with a clear testable hypothesis stated?

-Is the study design appropriate to address the stated objectives?

-Is the population clearly described and appropriate for the hypothesis being tested?

-Is the sample size sufficient to ensure adequate power to address the hypothesis being tested?

-Were correct statistical analysis used to support conclusions?

-Are there concerns about ethical or regulatory requirements being met?

Reviewer #1: See general comment to the Authors

Reviewer #2: Details must be provided of sampling methodology (e.g. when during the year, etc) and sampling strategy (type of sample [random, convenience, etc]; sampling frame, etc). As far as I can see the sampling was not representative so may be subject to bias – this should be discussed. The authors say it is a ‘geo-surveillance’ study but provide no information on whether the sample is representative or geographic risk. Some degree of precision needs to be provided for prevalence estimates (i.e. 95% CI) and information should be provided for sample size and justification for sample size. Saying the Bronx has a 67% prevalence of positive samples is difficult to interpret without some measure of precision.

Viability testing can be done by egg incubation which is probably a better approach than microscope alone. This requires some discussion as to how useful is microscopy alone.

Reviewer #3: (No Response)

**Results**

-Does the analysis presented match the analysis plan?

-Are the results clearly and completely presented?

-Are the figures (Tables, Images) of sufficient quality for clarity?

Reviewer #1: See general comment to the Authors

Reviewer #2: A useful addition to this manuscript would be to use the NHANES Toxocara seropostivity data (if available for these NYC areas which the authors refer to in the text) to map geographic risk and associate this with the findings of this environmental survey. This would thus link environmental contamination risk with actual human exposure. Surely risk of positive samples by area could be modelled statistically? Such modelling could control for factors such as socioeconomic level, etc.

Table 1 should provide more information showing number of samples and number of locations, proportions of samples positive from each location (parks vs. playgrounds, by microscopy, PCR, or either), whether any sample was positive at each location, etc.

Figure 3 indicates that qPCR correlates perfectly with egg count. Is this true?

Figure 5 should show medians rather than mean. One outlier for the Bronx biases the mean.

Reviewer #3: (No Response)

**Conclusions**

-Are the conclusions supported by the data presented?

-Are the limitations of analysis clearly described?

-Do the authors discuss how these data can be helpful to advance our understanding of the topic under study?

-Is public health relevance addressed?

Reviewer #1: See general comment to the Authors

Reviewer #2: Limitations need to be discussed more extensively (see comments above)

Reviewer #3: (No Response)

**Editorial and Data Presentation Modifications?**

Reviewer #1: See general comment to the Authors

Reviewer #2: The introduction could be cut significantly without affecting the message of the manuscript.

Reviewer #3: (No Response)

**Summary and General Comments**

Reviewer #1: The ms PNTD-D-19-01601 describes a study aiming at evaluating the environmental contamination of New York City with the eggs of the zoonotic nematodes Toxocara spp. The study has been carried out properly and it is well described, although I suggest some refinements to the Authors before considering it fully suitable for publication.

General comments

The concept that “infective Toxocara eggs are more predominant in poor socio-economical settings” is redundant throughout the ms and explained and discussed too much. I suggest the authors to give such an explanation of the higher presence in the Bronx rather than in the other boroughs once, in a detailed way, without reiterating it various times. The Introduction should also be reworded accordingly, as the ingestion of infective Toxocara eggs by a child poses relevant sanitary issues regardless the family income or the socioeconomic setting of the borough. A child may put in his/her mouth contaminated fingers in Central Park as also in the Bronx. Than, if some areas are more contaminated than others, this increases the risk but the sanitary importance and consequences are the same. Thus, the Study has the primary aim to evaluate the presence of contamination by Toxocara eggs in NYC and then (secondary objective) to compare its level in the five boroughs (see end of the Author summary).

Examples of repetitions: In the last three lines of the abstract this is reported twice, half of Page 3, first line of Page 4

I suggest the Authors to avoid wording like “we” “our”, etc, and rather use an impersonal style.

The full name of the parasites (Toxocara canis and Toxocara cati) should be reported only the first time the nematodes are cited, or at the begin of a sentence. All the other times , they should read T. canis or T. cati (abbreviating the genus). Please amend the text accordingly. 

Specific comments

Author summary

First line: add “respectively” after “cats”

Sentence from “Since” to “York City”: I would say that the need of such research relies on the sanitary impact of Toxocara for both humans and animals, rather than because humans are not definitive hosts of the parasite

Introduction

Half of page 3: ….questions still exist surrounding these enigmatic parasites (and not this enigmatic parasite): Toxocara canis and Toxocara cati are two distinct species.

Last three lines: Toxocara should be in italics (as also at the end of Page 9, and in other parts of the text, please check).

Methods and Results

The Methods section should contain details of the areas that have been sampled and the samples that have been examined. How the areas within the boroughs have been selected and the samples have been collected. How many samples per each single area. These details should not be present in the Results section, that instead should include only details on the positive samples. 

Delete “(The dog form of Toxocara)” at Page 8

First two lines of Page 9: cats are source of contamination, not vectors (this term has another meaning in veterinary parasitology). Also, dogs may carry T. cati egg after ingestion of cat faeces, thus the eggs are transported passively in dogs’ intestine and eliminated via the faeces. It is a passive transportation, not an infection as it seems by the sentence. Please reword for clarity.

Discussion

Last paragraph: Rodents, birds, rabbit are paratenic hosts, not accidental. It would also be interesting to add at least a brief mention on what other “larvated parasites” were found in the samples

Reviewer #2: A study potentially of interest to readers of PLoS NTDs providing up-to-date data on environmental contamination with Toxocara spp in NYC. There are significant weaknesses with respect to the information provided on sampling strategy and the data presented.

Reviewer #3: The aim of the study by Tyungu and co-workers was to give an overview about the contamination rate with eggs of the zoonotic agent Toxocara spp. in soil of playgrounds in New York City, wherefore soil samples from 91 different playgrounds allocated in all five boroughs of NYC were collected. The prevalence varied between different boroughs. The authors draw the conclusion that the income and therefore socioeconomic conditions might be a valuable explanation for the discrepancies between the boroughs. Furthermore, the authors discriminated between Toxocara canis and T. cati by qPCR, leading to the assumption, that playgrounds are predominately contaminated with eggs of T. cati. 

Overall, the manuscript deals with an interesting topic and provides a comprehensive insight into the contamination rate in New York City with Toxocara eggs. However, I have general concerns about the manuscript in its present form (e.g., data reliability if 100 g vs. 150 g soil was used – see comment below). Methods and results are partially superficial and further information on the study design, e.g. seasonal distribution of sampling, on the statistical analyses and molecular analyses is necessary before it may be recommended for publication. Please consider my comments and recommendations below to improve the manuscript.

Introduction:

The Introduction comprises important aspects for the study, but sometimes a clear structure is missing. Rearranging the chapter with a clear structure will definitively make it easier to follow and will help to enhance the understanding why the study is important. 

“Stool” is the term in human medicine, for animals the term “feces” is used in veterinary medicine.

It is now generally accepted that the third-stage larva (L3), not the second stage, is the infective stage (see Brunaska et al. 1995: Toxocara canis: ultrastructural aspects of larval moulting in the maturing eggs. Int J Parasitol. 25:683–90.)

Regarding reference 15: There are further, more recent, large-scale studies on the association of Toxocara seropositivity and neurocognitive deficits, which could be mentioned here like reference 5, or: Erickson, L. D., Gale, S. D., Berrett, A., Brown, B. L., Hedges, D. W., 2015. Association between toxocariasis and cognitive function in young to middle-aged adults. Folia Parasitol. (Praha) 62:048.

The general migration route of Toxocara larvae in paratenic hosts should be mentioned as the accumulation in different organs leads to different clinical aspects of the disease. 

I am not sure if the last sentence of the Introduction really reflects the study objectives. Contamination rates of public places give only an overview about a potential risk of infection and cannot automatically be extrapolated to the health disparities between different communities. 

Methods:

How was the sampling frequency at the sampling sites? Just once or at different time points during the study period?

Analysed samples weighed 100-150 g. This is a huge variation (plus 50% in case of 150 g vs. 100g) and may have influenced the chance of egg recovery and egg recovery rates, possibly leading to biased results. This needs at least to be mentioned in the Discussion. (e.g., were the Bronx samples those with the highest sample weights or were the 6 qPCR positive samples those with the highest weights?)

What was the concentration of Tween20 used for washing? What were the ratios of soil sample and flotation/washing solution?

Please give “g” instead of rpm throughout - rpm is not a useful unit because force varies with the radius of the centrifuge. Using g allows other researchers to replicate experiments.

Please use degree Celsius instead of Fahrenheit to be consistent with the rest of the manuscript.

Please give the sequence of the reverse primer(s). 

How was the DNA for the standard curve isolated from Toxocara eggs? With the same procedure used for the samples? And what egg numbers were used to construct the standard curve?

Statistical analyses: It is unclear what the difference between the two tests (Toxocara eggs by zip code and Toxocara eggs by individual boroughs) is, and why both a Kruskal-Wallis test and an ANOVA was conducted. The ANOVA does not compare prevalence, as egg count data was used (same as in the Kruskal-Wallis test?), please rephrase (prevalence would be presence/absence data). Were post-hoc tests conducted to clarify which boroughs differed significantly?

Please note that if sample sites were sampled more than once, this repeated sampling would need to be accounted for in the statistical analyses. 

An alternative statistical approach could be Generalized Linear (Mixed) Modeling, including the predictors borough, sampling month (for reasoning see below), and sample site as a random factor (in case of repeated sampling).

Results:

The Results section needs more detail, as several aspects remain unclear. 

Did the 91 sampling sites result in 91 samples or were these sites tested multiples times? And how was the overall prevalence for NYC?

It would be easier to follow if additionally the total numbers for each sampling area are given, e.g. 8/27 (29.6%) for Manhattan. 

Can the authors provide further data on the contamination intensity of the soil, like shown in Figure 5? Results of statistical analyses should also be mentioned, including which test produced which P-value. Which boroughs differed significantly – Bronx compared to all other ones?

Which other parasite stages were detected besides Toxocara eggs? – there is a brief hint in the Discussion, but it is of interest which parasites were detected as they might have zoonotic potential as well.

How was the success rate of the qPCR? Where the qPCR positives the samples with the highest egg counts in microscopy (please give counted egg numbers for the qPCR positive samples).

What is the meaning of the correlation to the 40-Cycle threshold? In the methods it is only mentioned that a sample was considered positive if there was detectable DNA at or before a cycle threshold of 38. Did additional samples become positive at Ct 40? What is the lower limit of quantification by the conducted qPCR extrapolated from the standard curve (this is an important information to draw conclusions on the possible co-existence of T. canis eggs in the qPCR positive samples or might explain failure to identify most of the samples and to detect T. canis at all)? Did the calculated egg count correspond to the egg count determined by microscopy? 

It is widely described that there are seasonal variations in the occurrence of Toxocara eggs in soil. Therefore, a breakdown how the egg burden may have varied between months would be an interesting additional information (if not possible for each borough, than for the whole city). This may provide an explanation why larvated eggs could only be detected in the Bronx. As known, embryonation accelerates with ascending temperatures and humidity. So, maybe the samples in the Bronx were collected after warmer periods? This is briefly mentioned in the Discussion, but without details on when each borough was sampled it remains unclear how much this has impacted the results. Multivariate analyses (see above) would allow to tease these effects apart.

Furthermore, the correlation between median income and percent positive places could also be tested statistically.

Discussion:

The limitations and the bias of the conducted qPCR, which was successful in six samples only, are well and plausible discussed. In all 6 samples T. cati DNA was detected (can co-contamination with T. canis be excluded – again, what was the sensitivity of the qPCR?). Even though T. cati is described to be the predominant species contaminating playgrounds in other cities, a samples size of 6 is too low to draw a conclusion for the whole city (consider also e.g. sensitivity of the qPCR; furthermore, you cannot exclude that all other positive samples contained T. canis eggs – be more cautious with your conclusions). 

How can the contamination rates be classified compared to other metropoles or cities with similar climatic conditions? There are several publications available e.g. Fakhri et al., 2018: Toxocara eggs in public places worldwide - A systematic review and meta-analysis. Environmental pollution, 242:1467-1475.

The human infection risk has been associated with a certain number of eggs per grams soil (e.g. Woodruff et al. 1981: Toxocara ova in soil in the Mosul District, Iraq, and their relevance to public health measures in the Middle East. Ann Trop Med Parasitol. 75:555-7). This is a relevant point when stating that infection risk was highest in the Bronx. Authors should give numbers of detected eggs in their samples in the manuscript, e.g. by including a table. 

Cleaning of play areas is mentioned - please include cleaning frequencies in public places in NYC in the MS.

Figures/Table:

Figure 1: The provided pictures of Toxocara eggs are not well focused, please provide better pictures if possible. 

Figure 2: As PLOS NTD is an international journal for scientists all over the world, please use the metric system. 

Figure 4: The income per zip code is not a series of consecutive measurements, therefore the points should not be connected via a line. 

Figure 5: It would be nice to illustrate which boroughs differed significantly – Bronx compared to all other ones?

Table 1: Please write 30.8 instead of 30.77 to be consistent with the other numbers. Are the +++ a semiquantitative measurement or just an estimation by the authors? This should be mentioned.

General comments:

Write species names such as Toxocara in italics throughout (check the text, figures and references).

The manuscript should be formatted according to the journals guidelines and checked for typos.

PLOS authors have the option to publish the peer review history of their article (what does this mean?). If published, this will include your full peer review and any attached files.

Reviewer #1: No

Reviewer #2: No

Reviewer #3: No

---

## [Decision Letter · Decision Letter 1]

23 Mar 2020

Dear Dr Mejia,

Thank you very much for submitting your manuscript "Neglected infection of poverty: Toxocara species environmental contamination in public spaces in New York City" for consideration at PLOS Neglected Tropical Diseases. As with all papers reviewed by the journal, your manuscript was reviewed by members of the editorial board and by several independent reviewers. The reviewers appreciated the attention to an important topic. Based on the reviews, we are likely to accept this manuscript for publication, providing that you modify the manuscript according to the review recommendations. 

Sincerely,

Celia Holland

Guest Editor

Christine Budke

Deputy Editor

Reviewer's Responses to Questions

**Key Review Criteria Required for Acceptance?**

**Methods**

-Are the objectives of the study clearly articulated with a clear testable hypothesis stated?

-Is the study design appropriate to address the stated objectives?

-Is the population clearly described and appropriate for the hypothesis being tested?

-Is the sample size sufficient to ensure adequate power to address the hypothesis being tested?

-Were correct statistical analysis used to support conclusions?

-Are there concerns about ethical or regulatory requirements being met?

Reviewer #3: (No Response)

**Results**

-Does the analysis presented match the analysis plan?

-Are the results clearly and completely presented?

-Are the figures (Tables, Images) of sufficient quality for clarity?

Reviewer #3: (No Response)

**Conclusions**

-Are the conclusions supported by the data presented?

-Are the limitations of analysis clearly described?

-Do the authors discuss how these data can be helpful to advance our understanding of the topic under study?

-Is public health relevance addressed?

Reviewer #3: (No Response)

**Editorial and Data Presentation Modifications?**

Reviewer #3: (No Response)

**Summary and General Comments**

Reviewer #3: I am comfortable with the changes made in response to my previous comments.

However, I do not agree with the title change in the revised manuscript. Leaving aside that none of the reviewers has asked for this change, human Toxocara infections are not only related to poverty, but more stringent to risk groups. For example, farmers, or veterinarians show significantly higher seropositivy than other professional groups. High prevalences in tropical countries are not only related to poverty, but also the climate, allowing eggs to become infective very fast and, additionally, enable survival in moist climates. Even though I agree that poverty (i.e. poor hygiene, but not necessarily low income in well developed countries) is one of various risk factors for humans to become infected, the study conducted by the authors is not related to the infection of poverty per se, it just shows higher egg contamination rates of sandpits in a borough with comparatively low income. If people in the Bronx are more frequently infected than in other boroughs cannot be answered or extrapolated from contaminated sandpits only.

PLOS authors have the option to publish the peer review history of their article (what does this mean?). If published, this will include your full peer review and any attached files.

Reviewer #3: No
---

## [Editor Report · Decision Letter 2]

24 Mar 2020

Dear Dr Mejia,

We are pleased to inform you that your manuscript 'Toxocara species environmental contamination in public spaces in New York City' has been provisionally accepted for publication in PLOS Neglected Tropical Diseases.

Best regards,

Celia Holland

Guest Editor

Christine Budke

Deputy Editor

---

## [Editor Report · Acceptance letter]

1 May 2020

Dear Dr Mejia,

We are delighted to inform you that your manuscript, "Toxocara species environmental contamination in public spaces in New York City," has been formally accepted for publication in PLOS Neglected Tropical Diseases.

Best regards,

Serap Aksoy

Editor-in-Chief

Shaden Kamhawi

Editor-in-Chief
